# Implementation of a PSO-Based Security Defense Mechanism for Tracing the Sources of DDoS Attacks [†]

**Hsiao-Chung Lin, Ping Wang * and Wen-Hui Lin**

Faculty of Department of Information Management, Kun Shan University, 710 Tainan, Taiwan;
fordlin@mail.ksu.edu.tw (H.-C.L.); linwh@mail.ksu.edu.tw (W.-H.L.)
* Correspondence: pingwang@mail.ksu.edu.tw; Tel.: +886-6-205-0545
† This paper is an extended version of our paper published in 2019 IEEE Eurasia Conference on IOT,
Communication and Engineering (IEEE ECICE 2019), National Formosa University, Yunlin, Taiwan,
3–6 October, 2019.

**Abstract:** Most existing approaches for solving the distributed denial-of-service (DDoS) problem focus on specific security mechanisms, for example, network intrusion detection system (NIDS) detection and firewall configuration, rather than on the packet routing approaches to defend DDoS threats by new flow management techniques. To defend against DDoS attacks, the present study proposes a modified particle swarm optimization (PSO) scheme based on an IP traceback (IPTBK) technique, designated as PSO-IPTBK, to solve the IP traceback problem. Specifically, this work focuses on analyzing the detection of DDoS attacks to predict the possible attack routes in a distributed network. In the proposed approach, the PSO-IPTBK identifies the source of DDoS attacks by reconstructing the probable attack routes from collected network packets. The performance of the PSO-IPTBK algorithm in reconstructing the attack route was investigated through a series of simulations using OMNeT++ 5.5.1 and the INET 4 Framework. The results show that the proposed scheme can determine the most possible route between the attackers and the victim to defend DDoS attacks.

**Keywords:** DDoS; particle swarm optimization; IP traceback; OMNeT++; attack route

## 1. Introduction

Although mobile devices and internet of things (IoT) devices with wireless sensing technologies for cloud appliances have improved the convenience of our daily lives, they also pose a threat to network attacks. The increasing number of things connected to the Internet poses new security worries, especially for the social internet of things (SIoT) in the extremely high complexity of the IoT environment [1,2]. Recently, vulnerable IoT devices were compromised by malicious users in order to perform distributed denial-of-service (DDoS) attacks. A denial-of-service (DoS) attack is characterized by an explicit attempt by attackers to prevent legitimate users of a service from using that service. Compared to a DoS attack, a DDoS attack is the most serious attack where the attack source is more than one, and often thousands of, unique IP addresses. In the DDoS attack, attackers first target any vulnerable part of your infrastructure. Then, attackers exploit the vulnerabilities of the services and insert hidden code, such that the malicious code can be protected from detection. After the attackers have compromised adequate hosts, they use the encrypted communication channels to attack the victims. Typically, DDoS attacks can be divided into two major categories: bandwidth attack and resource attack. A bandwidth attack simply tries to generate packets to flood the victim's network so that the legitimate requests cannot go to the victim's machine. A resource attack aims to send packets that misuse network protocol or malformed packets to tie up network resources so that resources are not available to the legitimate users any more [3].

Moreover, new routing protocols for IP-v6 inherited weak points that expose the vulnerability for threats and present information risks compared to existing IP-v4 protocols for dynamic network routing configuration. For example, the Routing Information Protocol next generation (RIPng) is one of the new routing protocols used in route discovery based on the distance-vector algorithm to compute the optimal route based on routing table entry (RTE), which selects the best route for each possible destination using distance as the main selection criteria. In the RIP v2, it embraced authentication to ensure that the routing updates are received from the preset sources. However, RIPng does not use authentication natively (RFC2080), but rather relies on IP-v6's Authentication Header (AH), one of the extension header types for authentication that allows an attacker to use a man-in-the-middle attack based on spoofing IPs to manipulate another provider's routers by using a compromised default gateway [4]. More concisely, the hackers misuse in obtaining the node information of the local area network (LAN) using five kinds of ICMPv6 messages for Dynamic Host Configuration Protocol (DHCP) operation in the route discovery process. In the following, a malicious user utilizes forging an NA (neighbor advertisement) message to generate a duplicated address detection (DAD) broadcast to initiate an attack on a single target by a multitude of compromised systems. To counter against cyber-attacks, the security managers use the IP traceback (IPTBK) scheme to periodically detect and identify the possible threats.

To defend denial-of-service attacks for network services, it typically involves the use of a combination of traffic classification, detecting the attack sources, and responding to the threat, aiming to block traffic that they identify as illegitimate and allow traffic that they identify as legitimate. As shown in Figure 1, a typical DDoS scenario is illustrated. In Figure 1, attackers use controlled zombies to generate a high volume of packets to flood the victim's network over the Internet so that legitimate requests cannot access the service. Further, the path reconstruction was processed using the IPTBK scheme to rebuild a trail along the route between the victim (food source) and attacker (nest) (e.g., route $V$-$R_6$-$R_7$-$R_8$-$A$), and the possible attack routes of DDoS attacks are indicated by calculating the relative probability of each feasible reachable route.

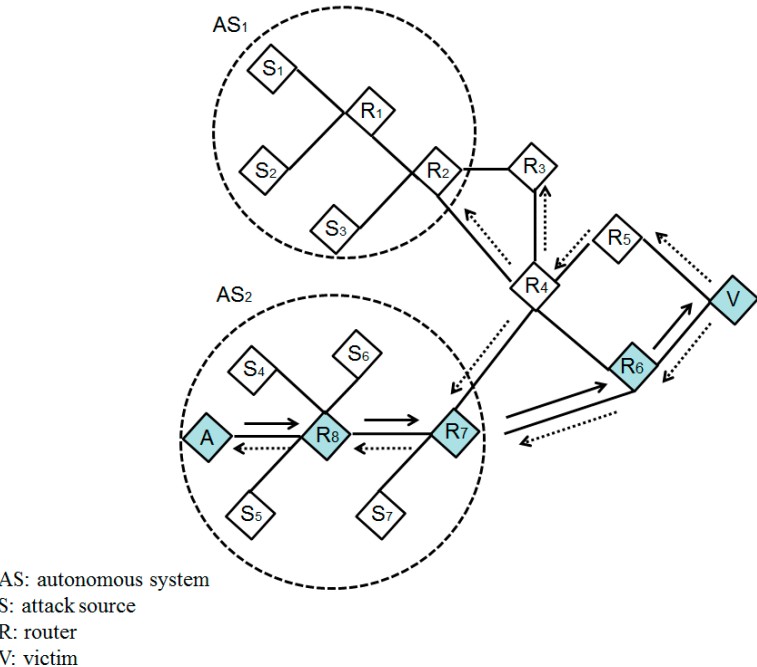

AS: autonomous system
S: attack source
R: router
V: victim

**Figure 1.** Attack route reconstruction in a distributed denial-of-service (DDoS) scenario.

Problem Definition

The problem of identifying the origin of an attack over the Internet is referred to as the IPTBK problem. Typically, the IPTBK problem involves collecting sufficient route information to determine all the possible routes between the attacker and the victim, given a constraint on both the quantity of routing packets collected and the computational time [5]. Solving the IPTBK problem is of crucial concern to information security management in detecting the origin of a malicious attack and bringing the perpetrator to court. As a result, many methods for reconstructing the attack route have been proposed in recent years.

Most existing methods for solving the IPTBK problem focus on DDoS attacks [6–8]. In a DDoS attack, a malicious user, referred to as a herder, utilizes a command and control (C&C) server to initiate an attack on a single target by a multitude of compromised systems. The resulting flood of incoming messages overwhelms the target, causing it to shut down and to deny access to legitimate users. For the defenders, knowing how to trace the attack source through use of analyzing possible traffic routes is a crucial problem. Basically, the IPTBK problem is not a trivial problem in large-scale network topologies, in which the aim is to attain a set of feasible solutions within polynomial-time under certain constraints, e.g., a limitation on the number of nodes within the topology. Problems of this type are most commonly solved using heuristic artificial intelligence algorithms such as ant colony optimization (ACO), genetic algorithms (GAs), PSO (particle swarm optimization), simulated annealing (SA), tabu search (TS), and evolutionary algorithms (EAs). Among these methods, PSO is a proven technique for solving a wide variety of combinational optimization problems, including vehicle routing problems (VRP) [9], energy efficient routing problems in wireless sensor networks(WSNs) [10], and cyber security [11].

Inspired by [12,13], the present study proposes a modified particle swarm optimization (PSO) scheme, designated as PSO-IPTBK, associated with a Source Path Isolation Engine (SPIE) mechanism [14] for solving the IPTBK problem subject to the constraint of minimizing both the number of routing packets required to reconstruct the route and the convergence time required for the reconstruction process. Also, the particle swarm updates the fitness function of feasible routes to assist the particle swarm of its neighbors to select the suitable route back to the attack source. The validity of the proposed algorithm is demonstrated by performing a series of experiments conducted by OMNeT++ 5.5.1 and the INET 4 Framework.

The primary contributions of this paper are:

- To identify the most probable attack route for assisting the defenders' design of DoS attack resistant systems, and a new tracing route-based IP traceback model with a revised PSO scheme by reconstructing the collected network packets to defend DDoS attacks.
- An improvement to the strategies of PSO in the optimal route searching process was proposed to prevent the PSO algorithm from converging prematurely to a local, sub-optimal solution in a big search space.
- The performance of the PSO-IPTBK algorithm in reconstructing the attack route was investigated through a series of simulations using a Monte Carlo model with OMNeT++ 5.5.1 and the INET 4 Framework to predict the accuracy of the proposed model.
- The experimental results revealed that the PSO-IPTBK accuracy is 98.33% for the attack scenarios in the experimental network (node = 24) and 94.64% for network topology (node = 40).

The remainder of this paper is organized as follows. Section 2 reviews related literature. Section 3 introduces the proposed analytical model to predict the possible attack routes with accuracy, which is based on the collected packets of DDoS attacks. Section 4 presents the simulations and analysis. Section 5 presents the conclusions.

## 2. Preliminary Work

This section overviews the use of two important issues, namely techniques for the IPTBK problem and PSO schemes in searching a global optimization algorithm for a possible attack source between the victim and the attack origin(s).

### 2.1. Techniques for the IP Traceback Problem

The IP traceback techniques are frequently roughly divided into two categories: (i) pro-active and (ii) passive. The former is an on-going traceback that needs to specify attack network flows from normal network flows immediately and typical schemes are input debugging, overlay network, and link testing. The latter filters and stores the partial path information used for identifying the attack sources after an attack, such as ingress filtering, probabilistic packet marking (PPM) [15], iTrace [16], deterministic packet marking (DPM) [7], and SPIE [14]. These IP traceback techniques are summarized in Table 1.

**Table 1.** Techniques for IP traceback. PPM: probabilistic packet marking; DPM: deterministic packet marking, SPIE: Source Path Isolation Engine.

| Approach | Features | Advantages | Disadvantages |
|---|---|---|---|
| Pro-active | | | |
| | · The basic idea is to perform attack source traceback during the attack. <br> · The well-known schemes include input debugging, overlay network, link testing, etc. | · Generally, pro-active schemes provide a suitable means of an on-going traceback. | · It is necessary to specify attack network flows from normal network flows immediately. <br> · It is assumed that all Internet Service Providers (ISPs) are cooperative during the attack, but in practice the ISPs are individually dominant in the relevant market. |
| Passive | | | |
| | · Can be only used for attack source traceback and identification after an attack. <br> · The well-known schemes are PPM, DPM, iTrace, and SPIE. | · Reduce significant storage overhead on routers. <br> · Generally, these schemes can reduce computation complexity of path reconstruction. | · ISPs of attack routes will cooperate to mark network packets, or the routers provide extra ICMP information. |

Compared with those of the IP traceback methods, by reducing the number of packets collected, such as PPM and SPIE, which have produced pretty good results, and are generally considered as the best solutions, have been used for this study. It implies that it is necessary to find an efficient scheme, such as PSO, ACO, and GA, to reduce computation complexity of route reconstruction from filtering partial path information.

### 2.2. Particle Swarm Optimization Algorithms

The PSO algorithm was first introduced by Kennedy and Eberhart in 1995 for simulating social behavior of bird flocking or fish schooling. In computational science, PSO is a computational method that globally optimizes a problem by iteratively trying to improve a candidate solution with regard to a given measure of quality via a particle swarm searching for the global minimum of a function.

In theory, PSO is a metaheuristic as it makes few or no assumptions about the problem being optimized and can search large spaces of candidate solutions. Compared to existing heuristic artificial intelligence algorithms, such as genetic algorithms, simulated annealing, and other global optimization algorithms, PSO could provide faster convergence and could find better solutions [13]. Essentially, the implementation of PSO is simple because PSO has no evolution operators, such as cross over and

mutation like GAs. Nevertheless, the main disadvantage of PSO is with its poor local search ability, i.e., it might cause an algorithm to converge to a sub-optimal solution.

In a later study, the performance of PSO was enhanced via three new search strategies, namely discrete PSO, constriction coefficient, and bare-bones PSO, which informed PSO to make PSO have a faster convergence and more reliability. More concisely, Shi and Eberhart (1998) [17] developed a boosted approach to change the particle swarm speed for the optimal route search within a complex solution space, designated as Inertia weight, in which Inertia weight $\omega$ is a proportional agent that is related with the speed of the last time. The bigger $\omega$ is, the bigger the PSO's searching ability for the whole is, and the smaller $\omega$ is, the bigger the PSO's searching ability for the partial. Generally, $\omega$ is equal to 1.

Suppose that each particle in PSO moves in the D-dimensional problem space with a velocity which is dynamically adjusted according to the moving experiences of its own and its neighbors. The location of the $i^{th}$ particle is denoted as $x_i = (x_{i1}, \ldots, x_{id}, \ldots, x_{iD})$ where $d \in [1, D]$, $_{i1}, _{iD}$, are the lower and upper bounds for $d^{th}$ dimension, respectively. The best previous position of the $i^{th}$ particle is saved and denoted by $P_i = (p_{i1}, \ldots, p_{id}, \ldots, p_{iD})$, which is also called $P_{best}$. The best particle among all the particles in the swarm is denoted by $P_{gbest}$. The velocity for the $i^{th}$ particle is denoted by $V_i = (v_{i1}, \ldots, v_{id}, \ldots, v_{iD})$, and is limited by $V_{max} = (v_{maxi1}, \ldots, v_{maxid}, \ldots, v_{maxiD})$, which is specified by the user.

As shown in Figure 2, the particle swarm optimization is based on changing the velocity and position of each particle toward its optimal locations $P_{best}$ and $P_{gbest}$, respectively. Notably, each particle has its memory to record the previous best position itself and the best position discovered by any particle in the swarm. That is, when the particle moves to a new position, then the fitness value of new position is re-calculated. If the fitness value of new position is better than that of the previous best position (i.e., $P_{best}$), the value of $P_{best}$ would be replaced by the new position updated by the particle's self-optimal experience. Similarly, $P_{gbest}$ would be replaced by $P_{best}$, if the fitness value of new position is better than that of $P_{gbest}$.

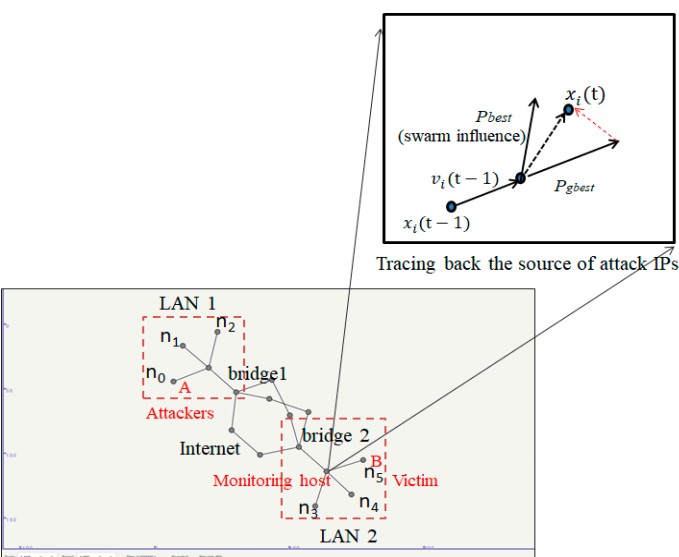

**Figure 2.** Attack route reconstruction in a DDoS scenario using a particle swarm optimization (PSO) scheme.

To avoid a particle being far away from the searching space, the speed of the particle created at its every direction is restricted at $[-v_{max}, v_{max}]^D$. In other words, stable speed of a particle in the swarm attracts a greater number of particles to follow. While the value of $v_{max}$ is too small, the solution might approach the local optimal position, and vice versa. To search the global optimal solution of PSO, the movement of each particle in the swarm is affected by different experiences, i.e., (i) individual

experience—the most optimal position during its movement (i.e., $P_{best}$), i.e., traced by the inertia of pervious velocity over a trail from the individual experience, however, this strategy might lead to the algorithm converging to a local optimal solution; (ii) group experience—each individual tries to emulate the position and velocity of the best of its neighbors (i.e., $P_{gbest}$), i.e., reinforce the particle's sight ability for direction searching, inspired by swarm wisdom that make the particles' search more flexible than that of the former.

Due to updating, the rule of PSO is deeply dependent on $P_{best}$ and $P_{gbest}$; the solutions to be searched might be confined to the local optimal solution. Angeline (1998) [18] introduced a scheme, namely "Selection", to solve this problem gradually to prevent premature convergence to a local sub-optimal solution. In general, swarm intelligence-based schemes provide an effective means of solving a range of complex optimization problems in the computer science and operations field, including artificial neural network training, job-shop scheduling problems, unmanned vehicle routing (UVR) problems, and data mining problems.

## 3. Proposed PSO Heuristic Models for IP Traceback Problem

In the proposed scheme, PSO is used as a global optimization algorithm for dealing with problems in which a best route of possible attack source between the victim and the attack origin(s) can be represented as a meta-optimization search problem.

### 3.1. Basic Concept

Assume that the IPTBK problem is to be solved using the PSO with a packet marking mechanism SPIE. In the solution procedure, the associated routers store partial route information during the attack process (i.e., record packet digests of sampled packets $p\%$). In the study, we used a small sampling rate $p$ (3.0%) to record minor storage and computational cost in practical networks. Then, the network intrusion detection system (NIDS) uses recursive lookup to reconstruct the attack route associated SPIE mechanism. The SPIE, the Source Path Isolation Engine, uses auditing techniques to support the traceback of individual packets while reducing the storage requirements. Notably, traffic auditing is accomplished by computing and storing 32-bit packet digests rather than storing the packets themselves in the SPIE. More detailed information regarding the format of 32-bit packet digests in the IP Header can be seen in [14].

Once the attack route information has been collected, the IPTBK problem is to be solved using a particle swarm comprising $m$ particles, and the position of each particle stands for the potential solution in $D$-dimensional space. In the solution procedure, each particle establishes a complete route between the attacker and the victim by choosing network nodes to visit according to the following three state change rules: (1) keep its inertia over a trail, (2) update the new position according to the locally best found position in history, and (3) possibly revise the new position of the particle swarm according to the most optimal position decided by swarm experience [19].

To simulate traceback using the proposed PSO-IPTBK scheme, the particle colony ($m$) represents the number of network packets collected. The route construction process is accomplished using a velocity-state change rule for position updating comprising two factors, i.e., $P_{best}$ and $P_{gbest}$. In solving the IPTBK problem using PSO, each particle swarm builds a route between the victim and the attacker by repeatedly applying velocity-state updating rules as Equations (1) and (2). As shown in Equation (1), the speed of each particle is updated by updating the velocity of each particle toward its optimal locations $P_{best}$ and $P_{gbest}$ at each time step in the particle swarm optimization process.

$$v_i(t + 1) = w_i \cdot v_i(t) + c_1 \cdot rand() \cdot (P_{best} - x_i(t)) + c_2 \cdot rand() \cdot (P_{gbest} - x_i(t)) \tag{1}$$

where $v$ and $x$ separately stand for the speed of the particle $i$ ($i = 1, \ldots, m$) at time $t$ and its position; $c_1$ and $c_2$ represent the acceleration constants, regulating the length when flying to most particles of the whole swarm and to the most optimal individual particle. The proper acceleration constants for $c_1$ and

$c_2$ can control the speed of the particle's movement. Typically, $c_1$ is equal to $c_2$ and they are equal to 2; rand()·represents it generating a random number located in [0,1]. In practice, $x_i(t)$ represents the traffic amount of previous node at time $t$ and the traffic amount of the new position of the node $x_i(t + 1)$ is then computed by the sum of $x_i(t)$ and the new velocity (i.e., throughput considering link state of next router) by Equation (2).

$$x_i(t + 1) = x_i(t) + v_i(t + 1). \tag{2}$$

The position updating rule of the particle in the space searching process was updated using Equations (1) and (2) continually until the particle $p$ traversed the most probable attack route. In the following, the performance of the PSO-IPTBK scheme was quantified using the following coverage percentage metric:

Coverage percentage(%) = (Average number of packets/attack path)/(Total number of routing packets) (3)

where the average number of packets on the attack route is computed as the total number of packets on the route divided by the routing distance (in hops). If the converged solution is not the true attack node, the average number of packets on the route is reset to zero and the search for the true route is resumed. The complete computational process for the IPTBK problem in revised PSO is illustrated as Figure 3.

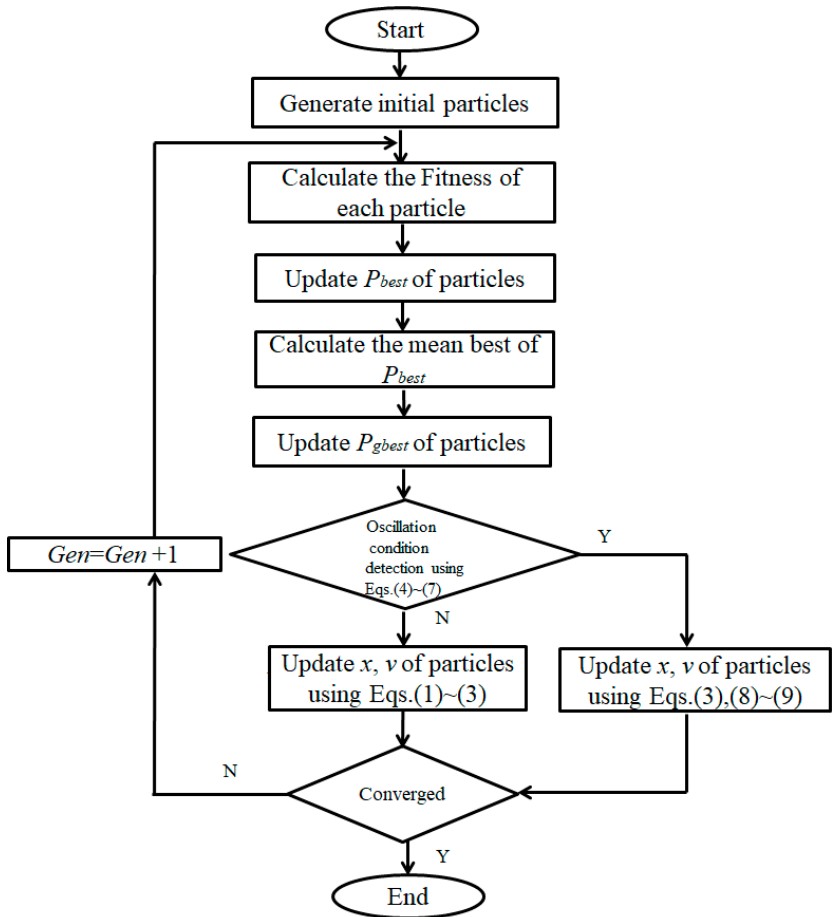

**Figure 3.** The resolution process of the revised PSO for IP traceback (IPTBK) problems.

Note that all of the network nodes visited by each particle swarm up to time $t$ are memorized in a tabu list to prevent the particle swarm from re-visiting the same node and removing a circle during the searching process. The process described above is repeated iteratively for a preset number of cycles. Finally, the particle swarms found the appropriate route that leads them back to the nest (i.e., finding the attack origins).

### 3.2. Application of PSO to the IP Traceback Problem

To solve IPTBK problems in *D*-dimensional space, NIDS calculate the fitness value of each feasible reachable route to determine a best route of possible attack source according to the network flow information using a PSO algorithm. Typically, two research objectives in implementing the optimization procedure are illustrated as follows: (i) to identify the most probable attack routes subject to the constraint of minimizing both the number of routing packets (cost), and (ii) the convergence time required for the reconstruction process in different network topologies. Assume that the attack route is to be a non-cyclic directed graph in order to ensure the convergence of the solution procedure.

Let the network topology be represented as a directed graph, $G = (x_i, e_i)$, where *X* represents a set of nodes, $x_i = \{x_{i1}, \ldots, x_{id}, \ldots, x_{iD}\}$; $x_s$ is a set of source nodes (i.e., attack sources); $x_d$ is a set of sink nodes (i.e., victims), and $e_i$ denotes the edges of the graph from node $x_i$ in *D*-dimensional space.

In the PSO model, two arbitrary nodes of network topology representing the attack source and the victim are chosen as end nodes of the traffic route. Solving the IPTBK problem using PSO exploits three fundamental characteristics of PSO schemes, namely information collection, cooperative learning, and distributed optimization. In the revised PSO scheme, a metaheuristic is proposed for increasing the diversity of the searched attack routes in the IPTBK problem. As discussed in the following, the algorithm is implemented using a four-step approach.

**Step 1:** Creation of network topology

The present study focuses on the security management aspect of network services. The service-oriented network topology is simulated and established for model analysis using OMNeT++ 5.5.1 and the INET 4 Framework. To examine the minimum quantity of packets required to reconstruct the attack route by the PSO-TPBK model to solve the IPTBK problem, various experimental network topologies will be constructed using a random graph generator based on Waxman's scheme [20].

**Step 2:** Determine the fitness value of feasible routes

In reconstructing the optimal routes between the victim and the attack source, each particle *i* in the swarm performs a complete search of all arcs at time *t* depending on its most optimal positions (i.e., $P_{best}$ and $P_{gbest}$) of the neighbors, as shown in Figure 3. To find the global optimal solution of the feasible routes, a random searching policy was given to re-construct the possible routes for discovering their route back to nest. The process is repeated iteratively in this way until the specified termination criteria are met.

Assume that there are multiple feasible routes $R_j$ among nodes (node$_i$, ..., node$_k$, ..., node$_j$) in the D-dimension space. The fitness value of each route is calculated to examine whether the particles will travel on these routes i.e., approaching in accordance with $P_{best}$ and $P_{gbest}$. In this study, the number of optimal routes between the victim node and the attacking nodes from collecting network marker packets (i.e., packet digests) are assessed by the fitness values of position for each route. That is, the fitness value of the new position in a route (*F*) is calculated to determine whether particles are included in a specific set that ensure these particles are along a route between the global best position and the current best position. When the distance between the new position and the best position discovered by any particle in the swarm $P_{gbest}$ is increasingly close, this usually implies the new position of particles $x_i$ belongs to a subset of best routes $R_j (x_i)$, as shown in Equations (4) and (5).

$$F = \| (x_i \subseteq R_j (x_i) \|, i = 1, \ldots, n; j = 1, \ldots, M \tag{4}$$

$$\text{If } \| D(x_i, P_{gbest})_{t+1} < D(x_i, P_{gbest})_t \| \text{ then } F = F + 1 \tag{5}$$

where $D\left(x_i, P_{gbest}\right)_j = \sqrt{\sum_{i=1}^{n}\left[x_i(i, j) - P_{gbest}(i, j)\right]^2}$, and where the fitness value of F represents a ratio of particles traversed on feasible routes $R_j (x_i)$. The larger the *F* value, the better the result. As shown in Equation (5), the route distance between the new position of the particle and the global best position $D (x_i, P_{gbest})$ is defined by the Euclidean distance function. Finally, the model accuracy is evaluated by the coverage percentage (%), which is the ratio of the average number of packets on an attack route to

the total number of routing packets. Also, the false rate of position updating (*FR*) for the fitness value of the new position is calculated as an evaluation criterion (as opposed to performance evaluation in general), that is, the ratio at which the particles are excluded towards the most optimal positions. Obviously, the lower the false rate, the better the prediction result.

$$FR = \sum \| (x_i \not\subset R_j(x_i) \|/M, \ i = 1, \dots, n; j = 1, \dots, M \tag{6}$$

where *M* represents the total number of feasible routes that the particles are along a route between the global best position in *D*-dimensional space in accordance with Equation (2).

**Step 3:** Route Traceback Processing

In solving the IPTBK problem using PSO, a particle colony builds the attack routes (i.e., a feasible route between the victim and the attackers) by repeatedly applying state updating rules as shown in Equations (1) and (2).

A revised updating rule of route construction.

A metaheuristic is proposed to increase the possibility of explored routes in a large search space, thereby preventing the PSO algorithm from converging prematurely to a local, sub-optimal solution. If competitive behavior of the PSO is used for searching feasible routes between two network nodes in D-dimension space, stability analysis of the PSO algorithm is not a trivial issue. In other words, there exists the problem of particle speed where it is not easy to determine the suitable value interval in a dynamic state. Actually, the smaller interval is one of the advantages of PSO. Sometimes, there is an oscillation condition in optimum solutions that causes an algorithm to converge to a local optimal solution. In order to make the particles jump out from one region to the bounded area, this study proposes a revised updating rule to approximate global optimization of the particle swarm optimization community.

Generally, the global best solution is updated with the static updating rule for speed, and is applied in Equation (1). While the distance between the particle position $x_i(t)$ and $P_{gbest}$ stay close to nearby, i.e., $\delta' \cong 0$, it may raise an oscillation condition in the optimal solution searching process. In an oscillation condition occurring in optimum solutions, the distance updating will be decreased at each step, which leads to the particles approaching a local sub-optimal solution, i.e., $\| D(x_i, P_{gbest})_{t+1} < D(x_i, P_{gbest})_t \| < \delta'$ for repeating it $\alpha$ *times* in the optimum solution searching process. As shown in Equation (7), we use a fitness value of feasible routes defined in Equation (5) to detect an oscillation condition occurring in optimum solutions as

$$\text{If } (\| D(x_i, P_{gbest})_{t+1} < D((x_{ii}, P_{gbest})_t \| < \delta') \text{ repeat } \alpha \text{ times.} \tag{7}$$

To accomplish the optimal solution in the searching process, it requires the fine-tuning of the particle position (i.e., Equation (2)) with an acceleration to make the particle jump out from one bounded region to the new region, as shown in Figure 4; the new position is updated by adding the ratio ($\rho$) of an interval distance of the solution space ($x_{max} - x_{min}$), as shown in Equations (8) and (9).

$$x_i(t + 1) = x_i(t) + \rho[(x_{max} - x_{min})], \tag{8}$$

$$\rho = (x_i(t) - x_{min})/(x_{max} - x_{min}) \tag{9}$$

where $\rho$ represents the acceleration of the detached area, and its value is [0, 1]. $\rho = 0$ means that is no oscillation (steady state condition that $x_i(t)$ is approaching $x_{min}$) in the optimum solution searching process, and the normal speed update was used, while $\rho = 1$ (unstable state with $x_i(t) = x_{max}$) means oscillation occurred, and the interval distance of ($x_{max} - x_{min}$) is used for making particles jump outside the bounded region. For example, suppose the boundary maximum of each dimension is (10, 10) for a 2-D area, the minimum of each dimension (*i, j*) is (1, 1), the original position for three particles is (2, 1), (3, 4), (5, 3), respectively. The position of particle swarm is represented by (2, 1, 3, 4, 5, 3). If $\rho = 1/2$, the updated position of particles is revised to (6.5, 5.5, 7.5, 8.5, 9.5, 7.5).

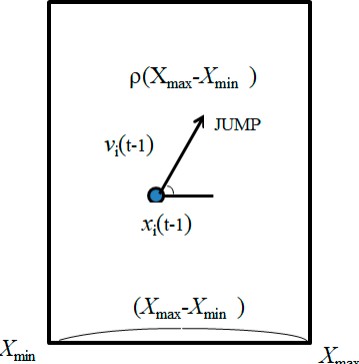

**Figure 4.** Improvement strategies of PSO in the optimal route searching process.

Repeating the searching process of feasible solutions, the updating rule would move the particle along a route towards the source and returning back in accordance with Equations (1)–(9). After the maximum number of iterations is reached, the updating process will stop if satisfactory fitness values of the updating position are attained, i.e., the optimal position of the particle swarm ($P_{gbest}$) remains static with no further improvement by performing the updating rule in Equation (7).

## 4. Results

In this section, the applicability of the proposed PSO-TPBK model is demonstrated by considering two examples of traceroute-based cloud security. The simulations were performed using a PC with an Intel Dual core CPU 3.0G, DDR3 2G of RAM, and the Ubuntu Desktop 18.04.3 LTS operating system associated with OMNeT++ 5.5.1 and the INET 4 Framework. OMNeT++ is an extensible, modular, component-based C++ simulation library and framework, primarily used for building network simulators which create a network of virtual hosts, switches, controllers, and links. The INET 4 Framework is an open-source model library for the OMNeT++ simulation environment. It provides protocols, agents, and other models for researchers and students working with communication networks. INET 4 is especially useful when designing and validating new protocols or exploring new or exotic scenarios. In the experiment, the file omnetpp.ini was used for configuring the simulation kernel, and entering the relevant parameter settings of the network topology. For example, the configuration of simulated network module) and the parameters of the recording network packet file in pcap format. In this experiment, a network intrusion detection system was constructed using the following three-step procedure, which involves: (1) creation of network topology, (2) data collection and threat analyses, and (3) reconstruction of attack routes, where workflow of security analysis is shown in Figure 3.

Case I: DDoS attacks using IoT devices

Step 1. Creation of network topology

Place 24 nodes at integer coordinates over a rectangle area of size 800 × 600 using OMNeT++, as shown in Figure 5. As shown in Figure 5, the simulated network topology consists of four local area networks (LANs). Simulated hosts and routers are configured using an XML file specified by the OSPFConfig parameter and the LANs are configured using omnetpp.ini and Ipv4ConFigure xm. Moreover, each of the four LANs have three host nodes, one switch node, one router node, and the relay nodes of the Internet. The attack sites are compromised IoT devices and are host1–host3, host7–host12, and the switch node is designated as switch1. The victim is a Web server (host4) in LAN2.

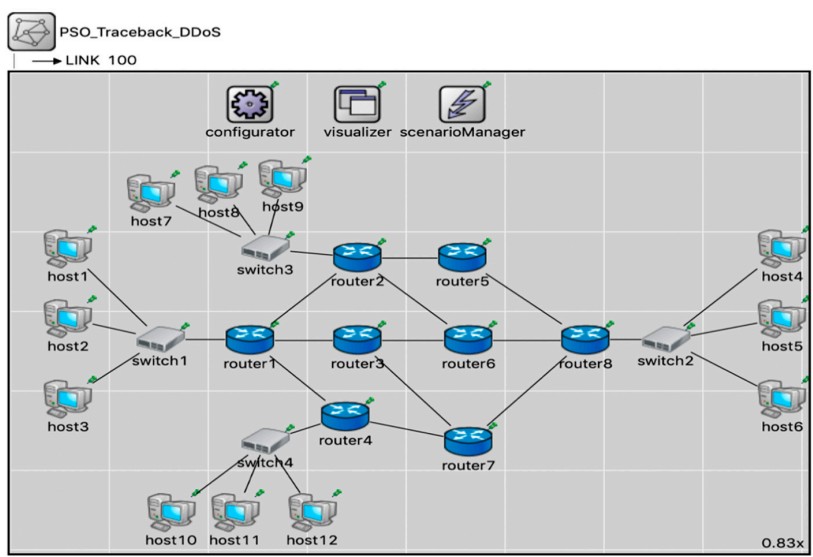

**Figure 5.** Simulated network topology (node = 24).

Step 2: Data collection and threat analyses

The attack routes were constructed using the following two-step procedure:

Step 2.1: Attack on victim

The attack nodes (IP address of 192.170.1.1–192.170.1.9) launch a high-rate DoS attack using UDP floods against the host4 in LAN2. The victim (IP address of 192.171.1.1) listens by default on port 4000. Six cycles of attacks are conducted in 120 s to generate routing information on the victim node for solving the IP traceback problem. A total of 1800 attack packets ($m$ = 1800) were sent to host4 using UDP floods. The average packet quantity of the visited node is counted as the basis of updating the number of particle and assisting particle swarms to trace back the sources of attacks by reconstructing route routes, as shown in Figure 6.

| Event# | Time | Relevant Hops Name | Source / ID | Destination / K | Protocol / Leng | Type | Length |
|--------|------|--------------------|-------------|-----------------|-----------------|------|--------|
| #6205 | 10.0002956 | switch1 --> router1 UdpBasicAppData-0 | 192.170.1.3:1025 | 192.171.1.1:5678 | UDP | 86 B |
| #6215 | 10.0003368 | router1 --> router2 UdpBasicAppData-0 | 192.170.1.1:1025 | 192.171.1.1:5678 | UDP | 86 B |
| #6225 | 10.00034464 | router1 --> router2 UdpBasicAppData-0 | 192.170.1.2:1025 | 192.171.1.1:5678 | UDP | 86 B |
| #6235 | 10.00035248 | router1 --> router2 UdpBasicAppData-0 | 192.170.1.3:1025 | 192.171.1.1:5678 | UDP | 86 B |
| #6245 | 10.00039368 | router2 --> router6 UdpBasicAppData-0 | 192.170.1.1:1025 | 192.171.1.1:5678 | UDP | 86 B |
| #6255 | 10.00040152 | router2 --> router6 UdpBasicAppData-0 | 192.170.1.2:1025 | 192.171.1.1:5678 | UDP | 86 B |
| #6265 | 10.00040936 | router2 --> router6 UdpBasicAppData-0 | 192.170.1.3:1025 | 192.171.1.1:5678 | UDP | 86 B |
| #6275 | 10.00045056 | router6 --> router8 UdpBasicAppData-0 | 192.170.1.1:1025 | 192.171.1.1:5678 | UDP | 86 B |
| #6285 | 10.0004584 | router6 --> router8 UdpBasicAppData-0 | 192.170.1.2:1025 | 192.171.1.1:5678 | UDP | 86 B |
| #6295 | 10.00046624 | router6 --> router8 UdpBasicAppData-0 | 192.170.1.3:1025 | 192.171.1.1:5678 | UDP | 86 B |

| Event# | Time | Relevant Hops Name | Source / ID | Destination / K | Protocol / Leng | Type | Length |
|--------|------|--------------------|-------------|-----------------|-----------------|------|--------|
| #6358 | 10.00067472 | switch2 --> router8 arpREPLY | 0A-AA-00-00-00-24 | 0A-AA-00-00-00-1C | ARP | 72 B |
| #6375 | 10.00073048 | router8 --> switch2 UdpBasicAppData-0 | 192.170.1.1:1025 | 192.171.1.1:5678 | UDP | 86 B |
| #6379 | 10.00073832 | router8 --> switch2 UdpBasicAppData-0 | 192.170.1.2:1025 | 192.171.1.1:5678 | UDP | 86 B |
| #6383 | 10.00074616 | router8 --> switch2 UdpBasicAppData-0 | 192.170.1.3:1025 | 192.171.1.1:5678 | UDP | 86 B |
| #6392 | 10.00078736 | switch2 --> host4 UdpBasicAppData-0 | 192.170.1.1:1025 | 192.171.1.1:5678 | UDP | 86 B |
| #6402 | 10.0007952 | switch2 --> host4 UdpBasicAppData-0 | 192.170.1.2:1025 | 192.171.1.1:5678 | UDP | 86 B |
| #6412 | 10.00080304 | switch2 --> host4 UdpBasicAppData-0 | 192.170.1.3:1025 | 192.171.1.1:5678 | UDP | 86 B |

**Figure 6.** Routing routes for high-rate DoS attacks from host1–host3.

Step 2.2: Data collection

Use the Tcpdump tool to collect the samples of network traffic flows on port 4000 of the victim where traffic flows are recorded in pcap format. Once attack flow packets have collected, a set of tools (scavetool) are applied to convert recording files to csv format for reconstruction of attack routes, as shown in Figure 7.

**Figure 7.** Convert network packet files to csv format.

Then, the defender uses the traceroute command to periodically collect the routing information of DoS attacks from routers and determines the route from a given source by returning the sequence of hops the packet traversed, as shown in Figure 8.

**Figure 8.** Routing trace—next hop.

In the following, the victim examines the number of packets collected from each attack node. For example, Router 8 received a total of 400 packets sent from host1 as shown in Figure 9. From Figure 9, this information indicates the attack source, which can be used for statistical data analyses later.

**Figure 9.** The victim examines the number of packets from each attack source.

Step 3: Reconstruction of attack routes

The routing information generated in Step 2.2 was used as the input dataset to the PSO model. Five important parameters in the PSO model are: (i) the population of the particle colony is set to the number of packets collected for DDoS attacks; (ii) execute the loop 10 times (generations) and update route searching rules by 500 iterations for each loop; (iii) initial value of $w_i$ (weighting factor) is 0.8; (iv) $c_1$ is equal to $c_2$ and they are set to 2.0 in Equation (2), respectively; and (v) $\rho = 1/2$ for the updating rule of route construction, if there is an oscillation condition occurring in the optimum process. Applying the searching rule to obtain optimal the PSO-TPBK solution, particles travel around all traits and back to the attack origins using the local and global updating rules given in Equations (1)–(9). Consequently, there are several possible attack routes to be discovered for model performance analysis.

Through the reconstruction of the attack route in Step 3, the backtracking of the attack end was performed by the revised PSO algorithm for the network topology, as shown in Figure 5. A total of 500 iterations were executed, 10 generations each time, and the execution results were consolidated and analyzed. The coverage percentage of experimental results is shown in Table 2. Select the first three columns in Table 2 as the possible attack routes for our experiment case with the choice of a minimum support threshold $t = 3\%$. From Table 2, it is revealed that the PSO-IPTBK accuracy is 98.33% for the network topology (node = 24) and the error rate is 1.67%.

**Table 2.** Possible attack route of DDoS attacks (node = 24).

| Attack Route | Packets Collected | Coverage Percentage (%) |
|---|---|---|
| router1(1)-router2(0)-router2(1)-router5(0)-router5(1)-router8(1) | 570 | 31.67% |
| router2(1)-router5(0)-router5(1)-router8(1) | 600 | 33.33% |
| router4(0)-router1(3)-router1(2)-router3(0)-router3(1)-router6(0)-router6(1)-router8(2) | 600 | 33.33% |
| router1(2)-router3(0)-router3(1)-router6(0)-router6(1)-router8(2) | 30 | 1.67% |
| Total | 1800 | 100.0% |

router$X(n)$ stands for router $X$ with $n^{\text{th}}$ network interface card, $n = 0,1,2, ..., n - 1$. For example, router2(5) represents the 6th network interface card of router 2.

The effect of the network size on the number of packets required to construct the attack path was also investigated. For similar test runs, a series of DDoS attacks for a simulated network topology (node = 40) were conducted to evaluate the convergence performance of the model as shown in Figure 10. In the experiment, the attacker was alternatively flooding the victim with packets originating from attack nodes in LAN1–LAN8 (except LAN2). A total of 8400 attack packets were sent in irregular bursts of packets to congest the link in 12 s. The coverage percentage of the experimental result is shown in Table 3. The results presented in Table 3 show that the PSO-IPTBK accuracy is 94.64% for the network topology (node = 40) and the error rate is 5.36%. Two experimental results demonstrated that the large-scale networks decrease the convergence performance of the PSO-IPBK scheme.

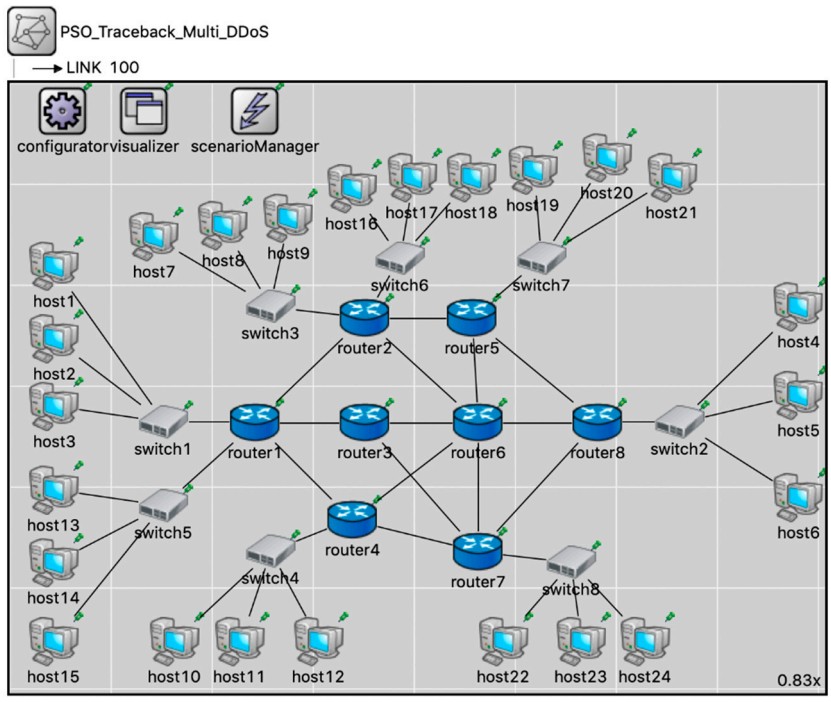

**Figure 10.** Simulated network topology (node = 40).

**Table 3.** Possible attack route of DDoS attacks (node = 40).

| Attack Route | Packets Collected | Coverage Percentage (%) |
|---|---|---|
| router7(4)-router7(2)-router8(3) | 1200 | 14.286% |
| router5(3)-router5(2)-router6(4)-router6(2)-router8(3) | 1200 | 14.286% |
| router2(3)-router2(1)-router5(0)-router5(1)-router8(1) | 1200 | 14.286% |
| router(0)-router1(1)-router2(0)-router2(1)-router5(0)-router5(1)-router8(1) | 1050 | 12.50% |
| router4(2)-router4(1)-router7(1)-router7(2)-router8(3) | 1050 | 12.50% |
| router1(0)-router1(2)-router3(0)-router3(1)-router6(0)-router6(1)-router8(2) | 300 | 3.571% |
| router4(2)-router4(3)-router6(3)-router6(2)-router8(3) | 150 | 1.786% |
| Total | 8400 | 100.0% |

Case II: Performance analysis for DDoS attacks using a Monte Carlo model

A series of experiments of performance analysis for DDoS attacks were conducted using a Monte Carlo model to predict the accuracy of the proposed model. Two hundred random attacks were simulated using a Monte Carlo model for particle colonies with $m$ = 1000, 4000, and 8000 particles, respectively, in order to generate routing information for each network node. Figure 11 reveals that for different particle sizes (i.e., $m$ = 1000, 4000, and 8000), the PSO-IPTBK performance decreased. In performing the simulations, the optimal route derived from the particle swarm for $m$ = 1000 was derived from those whose convergence was more rapid and performance efficient than that of particle swarm $m$ = 4000 and $m$ = 8000.

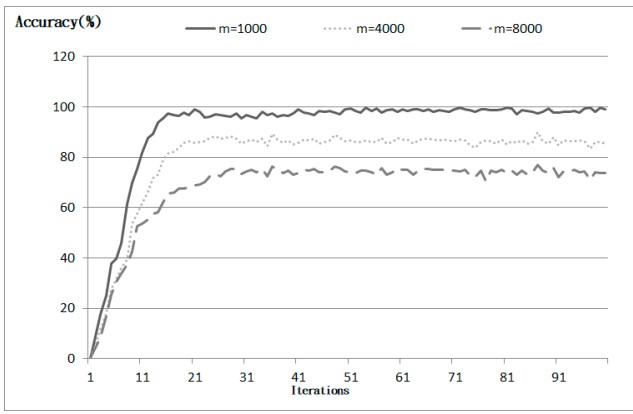

**Figure 11.** The coverage percentage of particles on attack routes with iterations.

## 5. Conclusions

This study has presented a PSO-based IP traceback model with a heuristic scheme for DDoS attacks, which allows defenders to reconstruct the most probable attack route. The application of the PSO scheme to the fake IP detection problem provides a reasoned approach for understanding the behavior of attackers. As a result, the proposed method improves the precision of the IP traceback solution. Moreover, the proposed scheme enables defenders to make an appropriate decision when tracing possible attack routes. The experimental results confirm the ability of the proposed scheme to detect the attack origin and the probable zombie over a trail. Obviously, collecting and classifying the network attack information is a non-trivial task. Thus, collecting the routing information from various attack events and consolidating this information within a database should be conducted strictly in advance. The scalability issue of complex network topologies in the IP traceback problem will be tackled in future studies. Moreover, the performance of the previously published IPTBK is further

extended to investigate and compare with the proposed PSO-IPTBK using a number of colony sizes, food sources, and iterations.

DDoS attacks utilize attack handlers and zombies to hide the identity of the real attacker.

**Author Contributions:** Conceptualization, P.W.; Methodology, P.W.; Resources, H.-C.L.; Analysis, H.-C.L.; Writing—Original Draft, H.-C.L.; Writing-Review and Editing, H.-C.L. and W.-H.L.; Funding Acquisition, P.W.; Software, H.-C.L.; Validation, H.-C.L. and P.W.; Visualization, W.-H.L.; Project Administration, P.W.

**Funding:** This research received no external funding.

**Conflicts of Interest:** The authors declare no conflict of interest.

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
