# Peer review of "Implementation of a PSO-Based Security Defense Mechanism for Tracing the Sources of DDoS Attacks†"

_computers, doi:10.3390/computers8040088_

Round 1

Reviewer 1 Report

My comments are given as blow.

In introduction section , there is is no graphical representation of DDOS attack. So, i suggest here to draw a diagram and explain the DDOS attack. The problem is based on IPTBK , hence you need to add one example based scenario . The motivation needs to be more highlighted. In preliminary work section, I suggest to add the summery of below studies.

1]Amin, F.; Ahmad, A.; Sang Choi, G. Towards Trust and Friendliness Approaches in the Social Internet of Things. Appl. Sci. 20199, 166. 

doi: https://doi.org/10.3390/app9010166

2]F. Amin, A. Ahmad and G. S. Choi, "Community Detection and Mining Using Complex Networks Tools in Social Internet of Things," TENCON 2018 - 2018 IEEE Region 10 Conference, Jeju, Korea (South), 2018, pp. 2086-2091.
doi: 10.1109/TENCON.2018.8650511

I didn't find the comparison table in related section section, hence i suggest to add one comparison table in this section and compare earlier studies to them.

In proposed solution i suggest to add one diagram that explain the working mechanism of your study. In addition, there is no bullets are used in the explanation of steps. hence you need to take care all of these.

All the best   

Author Response

Q1. In introduction section, there is is no graphical representation of DDOS attack. So, I suggest here to draw a diagram and explain the DDOS attack. The problem is based on IPTBK, hence you need to add one example based scenario. The motivation needs to be more highlighted.

Ans: Thanks for valuable comments. Add an illustrated diagram for attack route reconstruction in DDOS scenario (Fig.1) and also explain how to use IPTBK technique to rebuild the attack routes to identify the attack source. On Page 3, we also highlight two crucial contributions to enhance the reserch motives as follows.

The performance of the PSO-IPTBK algorithm in reconstructing the attack route was investigated through a series of simulations using a Monte Carlo model with OMNeT++ 5.5.1 and INET 4 Framework to predict the accuracy of proposed model  .

The experimental results revealed that the PSO-IPTBK accuracy is 98.33% for the attack scenarios in experimental network (node=24) and 94.64% for network topology (node=40) .

-----------------------------------------------------------------------------------------

Q2. In preliminary work section, I suggest to add the summery of below studies.

1] Amin, F.; Ahmad, A.; Sang Choi, G. Towards Trust and Friendliness Approaches in the Social Internet of Things. Appl. Sci. 2019, 9, 166. 

doi: https://doi.org/10.3390/app9010166

2] Amin F., Ahmad A. and Choi G. S., Community Detection and Mining Using Complex Networks Tools in Social Internet of Things, TENCON 2018 - 2018 IEEE Region 10 Conference, Jeju, Korea (South), 2018, pp. 2086-2091.
  doi: 10.1109/TENCON.2018.8650511

Ans:

On Page 1, add the security worries in social IoT environment to the first paragraph . Include two SIoT papers as important references in our manuscript.

-----------------------------------------------------------------------------------------

Q3. I didn't find the comparison table in related introduction section, hence I suggest to add one comparison table in this section and compare earlier studies to them.

Ans: On Page 3, add Sec. 2.1 ‘Techniques for the IP traceback problem’ and Table 1 to describe the method comparisons for IP traceback Techniques.

-----------------------------------------------------------------------------------

Q4. In proposed solution i suggest to add one diagram that explains the working mechanism of your study.

Ans: On Page 3, add the description of working mechanism regarding SPIE [14] in the first paragraph of Sec 3.1. Basically, we incorporated SPIE mechanism to cooperate with the associated routers and store partial route information during the attack process (i.e., record packet digests of sampled packets p%). In the study, use a small sampling rate p (3.0%) to record minor storage and computational cost in practical networks. Then the NIDS uses recursive lookup to reconstruct the attack route associated SPIE mechanism. The SPIE, the Source Path Isolation Engine, uses auditing techniques to support the traceback of individual packets while reducing the storage requirements. Notably, traffic auditing is accomplished by computing and storing 32-bit packet digests rather than storing the packets themselves in the SPIE. More detailed information regarding the format of 32-bit packet digests in the IP Header can be seen in [14]

---------------------------------------------------------------------------------

Q5. There is no bullets are used in the explanation of steps.

Ans: On Pages 5-11, add bullets for explanation of steps used in our model.

Reviewer 2 Report

Since the paper is an extended version of the published conference paper, I am more interested to see a section which highlights or justifies the new knowledge added in this manuscript. 

Author Response

Q1. Since the paper is an extended version of the published conference paper, I am more interested to see a section which highlights or justifies the new knowledge added in this manuscript. 

Ans: This manuscript was revised via several revisions by the published in 2019 IEEE Eurasia Conference on IOT, Communication and Engineering (IEEE ECICE 2019) and four major differences between this manuscript and conference paper of ECICE 2019 are listed as follows.

Add an illustrated diagram for a typical DDoS scenario as shown in. Fig.1 which shows that attacker uses controlled zombie’s to generate high volume of packets to flood the victim’s network over the Internet. Then, the victim use IPTBK scheme to rebuild the possible attack routes. Add a comparison table for techniques for IP traceback (see Table 1) and illustrates the feature, pros and cons among the methods. Also indicate two method, PPM and SPIE can efficiently reduce the number of packets collected and are generally considered as best solutions used for this study. Thus, PSO scheme is selected to predict the possible routes by reducing computation complexity of route reconstruction from filtering partial path information. The experiment was conducted by using a 24-node network topology with attackers in three LANs (Fig.5) in the conference paper of ECICE 2019. Considering the effect of the network size on the number of packets required to construct the attack path, we extended the experiment scale to a 40-node network topology with attackers in seven LANs (Fig.6) to examine the accuracy of proposed scheme by performing the pressure testing of DDoS attacks on the victim. In the revision, we added a series of simulations using OMNeT++ 5.5.1 and INET Framework to stimulate the scenarios of DDoS attacks, because the former provides more modular, component-based C++ simulation library to incorporate the revised PSO scheme into tracing route-based network simulations for PSO-IPTBK algorithm than that of ns-3. As a result, it increases the speed and extent of simulated experiments.

Round 2

Reviewer 1 Report

Authors has made all suggestions.